# Deciphering Reserve Mobilization, Antioxidant Potential, and Expression Analysis of Starch Synthesis in Sorghum Seedlings under Salt Stress

**DOI:** 10.3390/plants10112463

**Published:** 2021-11-15

**Authors:** Himani Punia, Jayanti Tokas, Virender Singh Mor, Axay Bhuker, Anurag Malik, Nirmal Singh, Abdulaziz Abdullah Alsahli, Daniel Ingo Hefft

**Affiliations:** 1Department of Biochemistry, College of Basic Sciences & Humanities, CCS Haryana Agricultural University, Hisar 125 004, Haryana, India; jiyaccshau@gmail.com; 2Department of Seed Science & Technology, College of Agriculture, CCS Haryana Agricultural University, Hisar 125 004, Haryana, India; virendermor@gmail.com (V.S.M.); bhuker.axay@gmail.com (A.B.); nirmalsingh@hau.ac.in (N.S.); 3Forage Section, Department of Genetics & Plant Breeding, College of Agriculture, CCS Haryana Agricultural University, Hisar 125 004, Haryana, India; satpal.fpj@gmail.com; 4Botany and Microbiology Department, Faculty of Science, King Saud University, Riyadh 11451, Saudi Arabia; aalshenaifi@ksu.edu.sa; 5Department of Food Sciences, University Centre Reaseheath, Reaseheath College, Nantwich CW5 6DF, UK; daniel.hefft@reaseheath.ac.uk

**Keywords:** α-amylase, germination, ion transporters, proline, reserve food, salinity, sorghum, starch

## Abstract

Salt stress is one of the major constraints affecting plant growth and agricultural productivity worldwide. Sorghum is a valuable food source and a potential model for studying and better understanding the salt stress mechanics in the cereals and obtaining a more comprehensive knowledge of their cellular responses. Herein, we examined the effects of salinity on reserve mobilization, antioxidant potential, and expression analysis of starch synthesis genes. Our findings show that germination percentage is adversely affected by all salinity levels, more remarkably at 120 mM (36% reduction) and 140 mM NaCl (46% reduction) than in the control. Lipid peroxidation increased in salt-susceptible genotypes (PC-5: 2.88 and CSV 44F: 2.93 nmloe/g.FW), but not in tolerant genotypes. SSG 59-3 increased activities of α-amylase, and protease enzymes corroborated decreased starch and protein content, respectively. SSG 59-3 alleviated adverse effects of salinity by suppressing oxidative stress (H_2_O_2_) and stimulating enzymatic and non-enzymatic antioxidant activities (SOD, APX, CAT, POD, GR, and GPX), as well as protecting cell membrane integrity (MDA, electrolyte leakage). A significant increase (*p* ≤ 0.05) was also observed in SSG 59-3 with proline, ascorbic acid, and total carbohydrates. Among inorganic cations and anions, Na^+^, Cl^−^, and SO_4_^2−^ increased, whereas K^+^, Mg^2+^, and Ca^2+^ decreased significantly. SSG 59-3 had a less pronounced effect of excess Na^+^ ions on the gene expression of starch synthesis. Salinity also influenced Na+ ion efflux and maintained a lower cytosolic Na^+^/K^+^ ratio via concomitant upregulation of *SbNHX-1* and *SbVPPase-I* ion transporter genes. Thus, we have highlighted that salinity physiologically and biochemically affect sorghum seedling growth. Based on these findings, we highlighted that SSG 59-3 performed better by retaining higher plant water status, antioxidant potential, and upregulation of ion transporter genes and starch synthesis, thereby alleviating stress, which may be augmented as genetic resources to establish sorghum cultivars with improved quality in saline soils.

## 1. Introduction

Salinity has become one of the major abiotic stresses that negatively affect plant growth and agricultural production worldwide [1]. According to the Food and Agriculture Organization (FAO), approximately 19.5% of the irrigated land and 2.1% of dry land is saline affected [2]. In addition, in arid and semi-arid regions, the salinization process occurs because of high evaporation and inadequate amounts of precipitation, leading to considerable leaching [3]. Sorghum (*Sorghum bicolor* (L.) Moench), a C_4_ plant, is the fifth most important cereal crop in the world, well adapted to various abiotic stresses with a wide range of nutritional values [4]. It is highly biomass productive, water-efficient, and widely cultivated in arid and semiarid tropics [5]. In recent years, it has transformed from being consumed as feed to raw material for the production of biofuels [6,7].

It is well known that climate change and environmental extremes induce and enhance the impact of abiotic stresses on plant fitness and performance [8,9]. Salinity has a significant impact on plant metabolism, generating ion toxicity, osmotic stress, mineral inadequacies, and physiological, biochemical, and metabolic alterations, eventually influencing plant growth, development, and productivity [10,11]. Accumulation of higher concentrations of inorganic cations such as sodium, magnesium, and calcium, and anions such as chloride and sulfate, disturb the membrane potential [12]. The high concentration of salts in the soil has a destructive effect on the cell membrane’s integrity, photosynthetic reactions, activation of several proteases, and absorption of nutrients [13]. Excess salts compromise the physiological and biochemical functions of plants, causing osmotic stress that results in disturbances of water conductance [2]. Tolerant plants use ions as an alternative to organic compounds for osmotic modification, which requires the synthesis of more energy (ATP) [14]. Under salinity stress, plants adapt by initiating multiple molecular and physiochemical changes, which results in modifications to metabolic pathways to reach a new homeostatic equilibrium [15]. Furthermore, defense through protective enzymes superoxide dismutase, catalase and peroxidase against salt-induced ROS over-generation and membrane lipid peroxidation is attributed to the protection of cellular membranes, which leads to salt tolerance [16].

One of the most significant processes in plants’ adaptation to stress conditions is the accumulation of reserve compounds (carbohydrates, proteins, and lipids) in seeds [17]. The function of these reserves is to provide energy for the formation of carbon skeletons within seedling tissues [18]. Salinity alters plant metabolism by impeding the mobilization of reserves and changing the embryonic axis membranes, but it is crucial to understand the reason for this process [19].

Salinity is also considered to activate alternative gene expression patterns that synthesize, degrade, or embellish metabolites from related pathways [20]. Monitoring metabolite patterns is critical for understanding the physiological and molecular responses of plants to salinity [21], as well as highlighting the functions of genes as crucial tools in functional genomics and systems biology for developing new breeding and screening strategies to enhance salt tolerance in crops [22].

The development of salt-tolerant crops has been a major objective of plant breeding programs to maintain crop productivity in semi-arid and saline areas has been a major objective of plant breeding programs [23]. This lack of information makes it imperative to study the damage caused by salinity during the mobilization of reserves in sorghum tissues since the germination and seedling establishment phases are vital for the success of the production. Understanding such mechanisms may contribute to the domestication as well as the genetic improvement of this species, making it profitable and competitive in semi-arid regions [24]. Screening of salt-tolerant sorghum genotypes is a practical approach for successful identification and plantation in saline-affected areas by the implementation of agronomic practices for high-yielding genotypes that includes soil preparation, variety selection, and plant management system. Thus, the objective of the present study was to examine the effect of salinity on the metabolism of reserve mobilization and partition of macromolecules in sorghum seedlings, through the quantification of antioxidants during the germination period for salt tolerance.

## 2. Materials and Methods

### 2.1. Plant Material and Treatments

The experiment was carried out in the Department of Biochemistry and Seed Science and Technology at CCS Haryana Agricultural University, Hisar, India. A preliminary experiment was conducted where 23 sorghum genotypes were screened for their tolerance behavior under different salt concentrations based on germination studies (Appendix A). Among the studied genotypes, SSG 59-3 and G-46 were identified as salt tolerant, CSV 44F as moderately salt tolerant, and PC-5 as a salt-susceptible genotype (Table 1). The germplasm was collected from the Forage Section, Department of Genetics and Plant Breeding, CCS Haryana Agricultural University, Hisar, India. To determine the salinity tolerance of sorghum genotypes, a Petri plate experiment was conducted in the laboratory at an ambient temperature of 28 ± 1 °C and relative humidity of 60 ± 5% to 81 ± 2% required for optimal growth of sorghum seedlings in three biological replicates. By using U.S saline laboratory staff solution [25], five different salt concentrations (60, 80, 100, 120, and 140 mM NaCl) were prepared. Distilled water was used as a control.

### 2.2. Phenotypic Trait Measurement

To assess the germination response to salinity, a germination experiment was conducted. A hundred seeds were used for each lot; germination paper was imbibed in different salt concentrations (60, 80, 100, 120, and 140 mM) and distilled water in a ratio of 2.5 times of paper dry mass. The plates were placed in a germination chamber at a constant temperature of 28 ± 1 °C. Prior to germination, seeds were surface sterilized in 0.1% HgCl_2_ solution for 60 s to prevent the fungal attack and then thoroughly washed with distilled water and air-dried. The sampling was carried out in triplicate. The germination test was carried out as per the ISTA procedure [26].

Germination percentage was calculated using the following formula:$$\mathrm{Germination}\;\left(\%\right)=\frac{\mathrm{Number}\;\mathrm{of}\;\mathrm{normal}\;\mathrm{seedlings}}{\mathrm{The}\;\mathrm{total}\;\mathrm{number}\;\mathrm{of}\;\mathrm{seeds}\;\mathrm{used}}\times 100$$

Seedling fresh and dry weight was recorded on the 7th day, and finally oven-dried at 80 ℃ for 48 h, then measured using an electronic analytical balance. A line was longitudinally traced in the upper third upon the humidified paper towel, where seeds were placed pointing the micropyle downward. The seedling’s length (root and shoot), which was considered normal, was determined at the end of the 7th day using a millimeter-gauged ruler. Seedling vigor indexes I and II were calculated as per the formula [27].
Seedling vigour index-I = Germination (%) × Seedling length (cm)
Seedling vigour index- II = Germination (%) × Seedling dry weight (mg).

### 2.3. Biochemical Analysis

Seven-day-old germinated sorghum seedlings were used for further biochemical analysis in three biological replicates. Lipid peroxidation (MDA) was estimated as TBARS [28]. Fresh seedlings were homogenized in 0.1% trichloroacetic acid (TCA) and centrifuged at 15,000 rpm for 15 min. The mixture of supernatant, 20% (*w/v*) TCA with 0.5% TBA, was heated at 95 °C for 30 min, and absorbance was recorded at 532 nm. The TBARS content was calculated using its extinction coefficient of 155 mM cm^−1^ and expressed in units (U).

Ascorbic acid was estimated in sorghum seedlings [29]. Seedlings were homogenized in 6% TCA and centrifuged at 10,000 rpm for 15 min. An aliquot was mixed with 2% 2, 4-dinitrophenyl hydrazine, one drop of 10% thiourea (in 70% ethanol), and kept in a boiling water bath for 15 min. 80% (*v*/*v*) H_2_SO_4_ was then added to the mixture at 0 °C (in an ice bath), and the absorbance was read at 530 nm. The quantity of ascorbic acid was determined from the standard curve of ascorbic acid (10–100 µg).

Total soluble carbohydrates were estimated by the method of [30]. Dried samples were extracted in 80% ethanol, and the filtrate was then collected. For estimation, diluted sugar extract, 2% phenol, and concentrated H_2_SO_4_ were added, and absorbance was measured at 490 nm. For proline determination [31], samples were homogenized in 3% sulphosalicylic acid, followed by centrifugation at 5000 rpm for 20 min. Ninhydrin reagent (1.25 g ninhydrin + 30 mL glacial acetic acid + 20 mL of 6 M phosphoric acid), acetic acid, and supernatant constituted the reaction mixture subjected to boiling to develop color. Afterward, toluene was added and colored in the non-aqueous phase and was read at an absorbance of 520 nm.

### 2.4. Physiological Indices

Electrolyte leakage/membrane injury was analyzed according to the method of [32]. Fresh tissue was immersed in deionized water and incubated for 3–4 h at room temperature. The conductance of effluxed electrolytes from decanted liquid was determined using a conductivity meter and designated as EC_a_. The samples were heated at 100 °C in a water bath for 30 min. After cooling, the conductance of the solutions was measured and defined as EC_b_ and expressed by the following formula:$$\mathrm{Electrolyte}\;\mathrm{leakage}\;\left(\%\right)=\frac{\mathrm{ECa}}{\mathrm{ECb}}\times 100$$

Relative water content (RWC) was measured from fresh samples [33]. Leaves were excised, weighed, and immediately placed in distilled water in diffused light for six h at room temperature. They were then dried in an oven at 70 °C for 72 h and RWC was calculated as follows:$$\mathrm{Relative}\;\mathrm{water}\;\mathrm{content}\;\left(\%\right)=\frac{\mathrm{Fresh}\;\mathrm{weight}\;–\mathrm{Dry}\;\mathrm{weight}}{\mathrm{Fully}\;\mathrm{turgid}\;\mathrm{weight}-\mathrm{Dry}\;\mathrm{weight}}\times 100$$

The osmotic potential was determined using a psychrometric technique (Model 5199-B Vapor Pressure Osmometer, Wescor Inc., Logan, UT, USA). Total chlorophyll content was extracted in fresh leaves using 80% acetone, then incubated in the dark at room temperature overnight and the absorbance was measured at 663 nm and 645 nm in a spectrophotometer (Beckman Coulter Inc., Fullerton, CA, USA). Using the absorption coefficients, the amount of chlorophyll was calculated [34].

Photochemical quantum yield (F_v_/F_m_) was recorded in intact seedlings using a chlorophyll fluorometer (OS-30p, Opti-Science, Inc., Hudson, USA) at mid-day. Initial (F_0_) and maximum (F_m_) fluorescence were recorded and variable fluorescence (F_v_) was derived by subtracting F_0_ from F_m_. Chlorophyll stability index (CSI) was estimated as described [35]. The CSI was calculated using as follows:$$\mathrm{CSI}\;\left(\%\right)=1-\frac{\left(\mathrm{Total}\;\mathrm{chlorophyll}\;\mathrm{of}\;\mathrm{heated}\;\mathrm{samples}\right)}{\mathrm{Total}\;\mathrm{chlorophyll}\;\mathrm{of}\;\mathrm{non}-\mathrm{heated}\;\mathrm{sample}}\times 100$$

### 2.5. Ion Profiling

The samples were dried and homogenized to a fine powder and subjected to acid digestion [H_2_SO_4_:HClO_4_ (9:1)] and the clear supernatant was analyzed by mass spectrometry (ICP-MS, Finnigan Element XR, Thermo Scientific, Bremen, Germany) to establish the inorganic cations and anions.

### 2.6. Enzyme Activities

#### 2.6.1. Ascorbate–Glutathione Pool

The antioxidative enzymes were determined in a homogenate of 1 g fresh tissue, prepared in 100 mM sodium phosphate buffer (pH 7.5) containing 0.25% (*v*/*v*) Triton X-100, 10% (*w/v*) polyvinylpyrrolidone, and 1 mM phenylmethylsulfonyl fluoride. Superoxide dismutase (SOD; EC 1.15.1.1) activity was determined by measuring the inhibition of NBT (nitroblue tetrazolium) reduction at 560 nm [36]. Catalase (CAT; EC 1.11.1.6) activity was assayed by monitoring the decomposition of H_2_O_2_ at 240 nm [37]. Peroxidase (POD; E.C. 1.11.1.7) activity was determined by the oxidation of pyrogallol (ε = 2.47 mM^−1^ cm^−1^) [38]. Ascorbate peroxidase (APX; EC 1.11.1.11) assay was based on the spectrophotometric monitoring of ascorbic acid oxidation (ε = 2.8 mM cm^−1^) [39]. Glutathione peroxidase (GR; EC 1.11.1.9) was assayed by monitoring non-enzymatic oxidation of NADPH (ε = 6.22 mM^−1^ cm^−1^) [40]. Glutathione reductase (GR; EC 1.6.4.2) was assayed by monitoring the oxidization of one mM NADPH per min (ε = 6.22 mM cm^−1^) [41].

#### 2.6.2. Alpha (α) Amylase and Protease Activity

α-amylase was estimated by the method described by Bernfeld et al. [42]. The germinating seeds (stressed and normal) were homogenized in sodium acetate buffer (100 mM, pH 4.7). Diluted 100 mM sodium acetate buffer (pH 4.7), diluted enzyme extract (1 mL), and starch solution (1% *w/v*) were added to the reaction mixture and incubated at 37 °C for 20 min. The enzymatic reaction was stopped by adding 3, 5-dinitrosalicylic acid and again incubated at 100 °C for 5 min. Then, 1 mL potassium tartrate (40% *w/v*) solution was added and mixed immediately by vortexing. The absorbance was recorded at 560 nm, and enzyme activity was expressed as μ mole min^−1^g^−1^ dry weight basis.

Protease activity was assayed in germinating seeds as per the method described by Lowry et al. [43]. The seeds were homogenized in 100 mM phosphate buffer at 4 °C. The reaction was initiated by adding casein (1% *w/v*) and incubated at 37 °C for 15 min. The reaction was terminated by adding trichloroacetic acid (5% *v*/*v*), and the resultant precipitate was removed by centrifugation at 12,000× g for 15 min. To the clear supernatant, Folin–Ciocalteau reagent was added and incubated in the dark for 30 min. The absorbance was recorded at 620 nm using a UV-visible spectrophotometer. The enzyme activity was expressed as μmole min^−1^ g^−1^ dry weight basis.

### 2.7. Starch and Protein Content

The starch content in the germinating seeds was estimated by using an anthrone reagent [44]. The tissue was homogenized in ethanol (85% *v*/*v*) and boiled for 15 min and centrifuged at 5000× *g* for 15 min. The absorbance was read at 630 nm and expressed as mg g^−1^ dry weight. The soluble protein content in germinating seeds was estimated as per Lowry et al. [43] using bovine serum albumin (BSA) as standard. The results were expressed as mg g^−1^ dry weight basis.

### 2.8. Semi-Quantitative Expression of Reserve Food Mobilizing Genes

The total RNA was extracted from seven-day-old sorghum seedlings using the Qiagen Plant Total RNA Miniprep Kit (Qiagen, Germantown, MD, USA). DNaseI was used to eliminate genomic DNA contamination in RNA samples. The extracted RNA was quantified using Picodrop (Picodrop Ltd., Cambridge, UK). The total RNA concentration was determined by absorbance at 260 nm, and the A260:280 ratio in the range of 1.8–2.0 was used for further analysis. Single-stranded cDNA was synthesized from the purified mRNA using an iScript cDNA synthesis kit (Bio-Rad Laboratories, Inc., Pleasanton, CA, USA). Normalization was achieved by dividing expression values by the total intensity (i.e., the sum of all expression values) of the given array. The relative expression levels of the genes were studied using 2X applied biosystems (ABI) Master Mix with gene-specific primers. The experiment was performed in three biological replicates for each sample and five technical triplicates. In a reaction volume of 25 µL, 2 µL of cDNA, gene-specific 0.2 µM of forward and reverse primer (0.5 µL each), Taq 2X master mix with standard buffer (12 µL), and nuclease-free water (10 µL) were used with the following thermal cycling conditions of 94 °C for 3 min followed by 30 cycles of duration at 94 °C for 30 s, annealing at 57 °C for 45 s, 72 °C for 60 s and 72 °C for 10 min in a thermocycler (Labnet MultiGene™ Mini PCR Thermal Cycler, Merck, Kenilworth, NJ, USA). The expression of each gene in different samples was normalized with *Actin-1* and *PP2A* (protein phosphatase 2A subunit A3) reference genes. The PCR product was analyzed on agarose gel (1.5%) containing 0.5 μg/mL SYBR Safe DNA Gel Stain (Thermo Fisher Scientific Inc., Waltham, MA, USA). The intensity of bands was quantified using Image J 1.51 K (National Institutes of Healt, Bethesda, MD, USA) (http://imagej.nih.gov/ij; 12 September 2021) densitometric software.

### 2.9. Primer Designing

Primer-BLAST software from NCBI was used for designing primers for the genes viz. α-amylase (*α-amy*), granule-bound starch synthase (*GBSS*), soluble starch synthase (*SS*), cysteine protease (*XCP1*), sodium proton antiporter (*NHX-1*), and vacuolar proton pyrophosphatase (*VPPase*) type-I. For normalizing the data, *Act-1* and *PP2A* were used as internal control. The primers were then custom synthesized from Integrated DNA Technologies, Inc. (Coralville, IA, USA), as listed in Table 2.

### 2.10. Statistical Analysis

The data were expressed as mean ± standard deviation (SD) (three replicates each). Two-way analysis of variance (ANOVA) was conducted to corroborate the significance of the main effects (genotypes and salinity) and their interaction on growth indices followed by post hoc comparison (Tukey’s test) at 5% level (*p* ≤ 0.05) using SPSS v23.0 software (SPSS for Windows, Chicago, IL, USA). Eigenvalues of the covariance matrix describe the proportion of total variance attributable to their respective principal components. The corresponding Eigenvectors of the principal components, representing the weight attributable to the measured traits for those principal components, were calculated using MINITAB 19 (Minitab Inc. LLC, State College, PA, USA).

## 3. Results

### 3.1. Effect of Salinity on Germination Parameters and Phenotypic Appearance Traits

Germination was observed after three days for all treatments (Figure 1). Germination variations among the studied sorghum genotypes are illustrated in Table 3. Seeds treated with distilled water germinated normally (79–95%), but their growth was adversely affected when treated with saline water more remarkably at 120 (36% reduction) and 140 mM NaCl (46% reduction) in all four genotypes (Figure 2). Amongst the available germplasm and screened genotypes, only SSG 59-3 (54–96%) and germplasm G-46 (49–94%) performed better at all the salinity levels. Results revealed that the decrement of germination due to salinity was lesser in SSG 59-3 (54.7%) and G-46 (48.5%), even at a higher salt concentration (140 mM), whereas PC-5 and CSV 44F were the poorest in performance (Table 3). Highly significant interaction effects were recorded between genotypes and salinity levels on root length and shoot length (*p* ≤ 0.05). In all the genotypes, shoot length was significantly (*p* < 0.05) reduced with increasing salt concentration (Table 3). However, SSG59-3 (11.56 cm) and G-46 (10.59 cm) had higher shoot lengths, whereas CSV 44F (7.93 cm) and PC-5 (3.21 cm) had the lowest at 140 mM NaCl. The decrease in shoot length was slightly smaller in SSG 59-3 (12.07 cm) and higher in PC-5 (5.35 cm) compared with their controls (22.5 cm and 14.3 cm, respectively). In the present study, the highest level (140 mM) of salinity had a more pronounced effect on root length to shoot length as roots were directly exposed to the salt solution. The data indicated that a reduction in water availability had adversely affected the root length of all genotypes. As with the shoot length, the cultivar’s root length exhibited the same trend under the same conditions (Table 3).

A significant effect of saline water on different genotypes was observed at 120 and 140 mM NaCl. Salinity stress drastically decreased the seedling vigor of screened sorghum genotypes. The growth rate of shoots controls the varietal difference in that lower shoot growth is associated with a higher level of NaCl in the leaves. The leaves of sorghum seedlings have accumulated with a rise in saline stress, accounting for 47.9~72.7% of total dry weight. The interaction between genotypes and salinity levels at the germination stage significantly affected the seed vigor index (Table 3). Results revealed that seedling vigor I and seedling vigor II were at a maximum in SSG59-3; in contrast, CSV 44F performed very poorly among all the screened genotypes when considered as salt-tolerant and salt-sensitive genotypes, respectively.

### 3.2. Physiological Responses of Sorghum Seedlings

The osmotic potential, total chlorophyll content, and electrolyte leakage varied significantly among salinity levels at *p* ≤ 0.05 (Table 4 and Table 5). In general, osmotic potential declined progressively with the increasing salinity levels (Table 4). Osmotic potential showed an upward trend and maximum in salt-sensitive genotypes at 140 mM NaCl, whereas it showed the minimum in tolerant genotypes (Table 4). Values of osmotic potential became more negative in PC-5 and CSV 44F as compared with SSG 59-3 and PC-5, i.e., −1.56 MPa, −1.62 MPa, −1.38 MPa and −1.42 MPa, respectively, at 140 mM NaCl. Total chlorophyll content was significantly affected by an increment of salinity levels (*p* ≤ 0.05) (Table 4). Salinity reduces total chlorophyll synthesis, the maximum and minimum total chlorophyll content was, respectively, in SSG 59-3 (1.77 mg/g DW) and CSV 44F (1.41 mg/g DW) compared with their controls, 3.32 and 2.75 mg/g DW, respectively, which showed that higher salt concentrations could ramp up chlorophyll decomposition in the body and reduce the photosynthetic efficiency of sorghum plants (Table 4). Membrane injury showed an increasing trend with increasing levels of salt stress from the control to 140 mM NaCl (Table 5). Maximum leakage was estimated at 120 and 140 mM NaCl in sorghum genotypes, but more leakage was seen in CSV 44F, i.e., 53.78% and less in SSG 59-3, i.e., 46.57%, compared with the control (23.32 and 16.54%, respectively).

Relative water content (RWC) was relatively low under salt stress compared with the control conditions in all the sorghum genotypes (Figure 2a). The reduction in RWC under salt stress was much higher in PC-5 (25.2%) than SSG 59-3 (7%) and G-46 (12%) at 100 mM NaCl. SSG 59-3 maintained RWC under increasing salt concentrations up to 100 mM NaCl, but at 140 mM, there was a significant decrease in RWC. A significant difference in photochemical quantum yield (F_v_/F_m_) was observed within the genotypes and treatments (Figure 2b). SSG 59-3 showed more quantum yield followed by G-46 and CSV 44 F than PC-5. Percentage reduction was higher in PC-5 (54% and 62%) and lower in SSG 59-3 (20% and 28%) in G-46, i.e., 24% and 32% at 100 and 120 mM NaCl, respectively. At 140 mM, the reduction was significantly higher in all genotypes. Up to 100 mM, all sorghum genotypes performed better. However, at 120 mM, the reduction was more, which drastically reduced the photosynthetic rate and conferred that 100 mM was under the physiological tolerance range of the crop. Salt stress significantly reduced the chlorophyll stability index (CSI) (%) of sorghum genotypes at *p* < 0.05 (Figure 2c). The reduction was more eminent in genotype PC-5 (65.3%) trailed by CSV 44F (45.2%), G-46 (28.2%), and lower in SSG 59-3 (22.9%). The decline in chlorophyll stability in terms of loss of chlorophyll was higher in PC-5 (85.3%) and lower in SSG 59-3 (30.2%) at 120 mM NaCl, suggesting the tolerance behavior of SSG 59-3.

### 3.3. Antioxidant Potential

MDA (lipid peroxidation) content significantly increased with a high salt environment (*p* ≤ 0.05) (Figure 3a). MDA content increased due to the accumulation of reactive oxygen species (ROS) synthesized under stress conditions. At 140 mM, MDA increased in salt-susceptible genotypes (PC-5: 0.300 and CSV 44F: 0.241 µmole/g FW), thus indicating an increase in MDA content. In contrast, salt-tolerant genotypes (SSG 59-3: 0.226 and G-46: 0.235 µmole/g FW), on the other hand, did not exhibit this increase in lipid peroxidation, indicating their tolerance towards salinity. Changes in ascorbic acid content in salt-tolerant and salt-susceptible genotypes due to salinity are shown in Figure 3b. Ascorbic acid content increased significantly in all the genotypes under stress. However, the percent increase was found to be higher in tolerant genotypes, i.e., 3.94 µmoles/g.FW in SSG 59-3 and 3.46 µmoles/g.FW in G-46, whereas PC-5 (1.71 µmoles/g.FW) and CSV 44F (1.45 µmoles/g.FW) had the minimum concentration of ascorbic acid. On revival, the ascorbic acid content continued to increase in all the genotypes; however, an increase was significant in tolerant genotypes. The proline content in fresh sorghum seedlings showed a statistically significant difference between control plants and other treatment groups (*p* ≤ 0.05). Salinity stress stimulated the accumulation of proline as a compatible osmolyte (Figure 3c). In comparison with the control, the proline content increased at increasing salinity levels. SSG 59-3 (55.0 µg/g.FW) possessed maximum proline indicating its tolerance behavior, whereas PC-5 (5.37 µg/g.FW) and CSV 44F (4.30 µg/g.FW) had the lowest at higher salt concentration. Total soluble carbohydrates act as an osmolyte inside the plant cell, and their accumulation increases the resistance against stress conditions. The total soluble carbohydrate content was significantly enhanced with increasing salt concentrations (*p* ≤ 0.05) (Figure 3d). Tolerant genotypes accumulated maximum soluble carbohydrates (SSG 59-3: 0.466 and G-46: 0.431 mg/g.DW), whereas sensitive genotypes accumulated less (PC-5: 0.366) and CSV 44F (0.335 mg/g.DW) at 140 mM. In this study, with the increasing saline stress, proline and soluble carbohydrate content in sorghum seedlings also increased to mitigate oxidative stress (Figure 3c,d).

### 3.4. Accumulation of Inorganic Cations and Anions

Inorganic cations (Na^+^, K^+^, Mg^2+,^ and Ca^2+^ ions) were significantly affected by salinity (*p* ≤ 0.05). The concentration of Na^+^ and K^+^ ions of sorghum seedlings changed drastically under saline stress. Na^+^ ion in sorghum leaves sharply increased with each salt concentration (Figure 4a). However, K^+^ ion decreased with the increasing saline stress (Figure 4b). The Mg^2+^ ion also reduced significantly (Figure 4c). The response to Ca^2+^ ion accumulation was similar to the Mg^2+^ (Figure 4d). Inorganic anions (Cl^-^ ion, and SO_4_^2−^ ion) were also significantly affected by salinity (*p* ≤ 0.05). The two anions showed the increasing trend of increasing salinity, whereas the accumulation of Cl^-^ ions was higher than the SO_4_^2−^ ion (Figure 4e,f). In the present research, the Na^+^ ion increased rapidly with increased saline stress. Conversely, K^+^ ion decreased. With the increasing saline stress, the Mg^2+^ ion and Ca^2+^ ion of sorghum seedlings decreased significantly. The Cl^-^ and SO_4_^2−^ anions showed a similarly increasing trend of increasing salinity, whereas the accumulation of Cl^-^ ions was considerably higher than the SO_4_^2−^ ion.

### 3.5. Reserve Food Mobilization in Sorghum Seedlings

Under stressed conditions, there is rapid mobilization of reserve food to provide energy for the synthesis of metabolites. At 120 and 140 mM NaCl, starch started to mobilize rapidly, and maximum mobilization was observed in SSG 59-3 and G-46 (Figure 5a), whereas in PC-5, the starch mobilization was slower. Its content remained higher at high salt concentration. Similarly, the soluble protein content in germinating seeds increased with salt concentration (Figure 5b). SSG 59-3 synthesized maximum proteins after exposure to salt treatment, whereas it was slower in PC-5. Thus, germinating seeds of SSG 59-3 showed higher starch mobilization and enhanced de novo protein synthesis at higher salinity levels.

### 3.6. α-amylase and Protease Activity in Germinating Seeds

The present study aimed to correlate the mobilization of starch and protein with α-amylase and protease activity. With salt stress exposure, the activity enhanced significantly with maximum activity in SSG 59-3 followed by G-46, whereas the minimum was observed in PC-5 at 120 mM (Figure 6a). At 140 mM, lower α-amylase activity was recorded in all genotypes due to inhibitory effects of Na^+^ ions. SSG 59-3 induced comparatively higher α-amylase activity than the control, but considerably lower at 140 mM. Similar to α-amylase activity, protease activity was also affected by high NaCl concentration in the same manner (Figure 6b). Protease activity was maximum in SSG 59-3 and G-46, and the lowest protease activity was recorded in PC-5. Furthermore, at 140 mM, enzyme activity had a marked falloff in all the genotypes. These data suggest that genotype SSG 59-3 significantly induced starch and protein mobilization by enhancing the α-amylase and protease activity in the germinating seeds.

### 3.7. Ascorbate–Glutathione Pool

Mild or severe salinity stress increased SOD activity in a concentration-dependent manner (Figure 7a). The percentage increase in SOD activity was a maximum in SSG 59-3 (51%) followed by G-46 (46%), considering them as salt-tolerant genotypes at 100 mM. In contrast, the increase was less in CSV 44 F (34%) and PC-5 (19%), suggesting them as salt-susceptible genotypes. The SOD activity was more pronounced at higher salt concentrations, i.e., at 140 mM, with an increase of 62% in SSG 59-3, 54% in G-46, and 32% in PC-5. Differential responses of CAT activity (Units mg^−1^ protein) are shown in Figure 7b. At 35 DAS, a significant increase in CAT activity was observed in SSG 59-3 (46%) followed by G-46 (39%), and a slight increase in CSV 44F (31%) and PC-5 (22%) at 100 mM. At 140 mM, the percent increase in CAT activity was higher in SSG 59-3 (62%) and G-46 (54%) as compared with PC-5 (32%). An increase in POD activity was observed in all sorghum genotypes (Figure 7c). The increase in POD activity in SSG 59-3 was 38% and 64%, 31% and 58% in G-46, whereas CSV 44F (29% and 45%) and PC-5 (22% and 29%) had a slight increase at 120 mM and 140 mM, respectively. APX activity increased significantly in SSG 59-3 (55%) and G-46 (48%), whereas in CSV 44F (34%) and PC-5 (26%), the increase in APX activity was slightly less at 100 mM NaCl (Figure 7d). Further increases in salt concentration, i.e., at 140 mM, enhanced the APX activity, more in SSG 59-3 (65%) and G-46 (61%) and less in CSV 44F (41%) and PC-5 (23%). Differential response of GPX activity (units mg^−1^ protein) showed a significant increase (Figure 7e). A significant increase in GPX activity was observed in salt-tolerant genotypes SSG 59-3 (45%) followed by G-46 (39%), with a slight increase in the salt-sensitive genotype PC-5 (24%) 100 mM. At 120 mM, the GPX activity was higher in SSG 59-3 (58%) compared with PC-5 (33%). Salt stress increased the GR activity in both the tolerant and susceptible sorghum genotypes (Figure 7f). However, the increase was found to be higher in tolerant genotypes. In the SSG 59-3 and G-46 genotypes, GR activity increased by 48% and 41% compared with CSV 44F and PC-5, where the percent increase was 32% and 21% at 140 mM. Among the studied genotypes, PC-5 showed the lower, whereas SSG 59-3 had the higher GR activity under salinity conditions.

### 3.8. Semi-Quantitative Gene Expression of Reserve Food Mobilizing Genes

In this study, we constructed the expression profiles of the coding genes for α-amylase (*α-amy*), granule-bound starch synthase (*GBSS*), soluble starch synthase (*SS*), and cysteine protease (*XCP1*), sodium proton antiporter (*NHX-1*), and vacuolar proton pyrophosphatase (*VPPase-I*). Using an RT-PCR assay, normalized expression levels of *α-amy*, *GBSS*, *SS*, *XCP1*, *NHX-1*, and *VPPase* were obtained for each genotype at 100 and 120 mM NaCl (see Appendix A). The appearance of a single band showed their amplification products (see Appendix A for corresponding profiles). SSG 59-3 displayed higher expression at 120 mM for *SS* (Figure 8a) and *GBSS* (Figure 8b), representing the adaptive behavior at higher salt concentrations; in contrast, for salt-sensitive PC-5, the relative abundance of *SS* and *GBSS* was relatively low. For *XCP1*, the relative expression peaked at 120 mM NaCl in SSG 59-3 (Figure 8c), whereas in PC-5, the expression was less, and for *α-amy*, the relative expression was higher at 100 mM (Figure 8d), whereas in PC-5, the lowest expression was observed for *α-amy*. In SSG 59-3, the higher expression of *NHX-1* (Figure 8e) and *VPPase-I* (Figure 8f) under high Na^+^ ions indicated its tolerance behavior to exclude the toxic effects of Na^+^ ions, whereas in PC-5, the up-regulation of these genes was lower. The expression was greater at 100 and 120 mM salt concentrations. The expression level of *VPPase-I* and *NHX-1* peaked at 100 and 120 mM salinity. *Act-1* (Figure 8g) and *PP2A* (Figure 8h) were constitutively expressed as stable under all salinity treatments. The higher expression of ion transporter genes under salinity indicates that SSG 59-3 may be utilized as a salt-tolerant crop due to its better genetic and agronomical traits. The detection of salt-tolerant genes could be adding a molecular base to identifying salt-tolerant genotypes in sorghum.

### 3.9. Principal Component Analysis

Considerable descriptive data regarding sorghum germination and adaptability were collected. Results of the initial PCA revealed that 92% of the variation among the lines was attributable to the first three principal components, nearly 100% to the first four principal components (Table 6 and Table 7). Data represented in a scree plot, loading plot, biplot, and score plot graphs show a relatively uniform distribution of the first three principal components for these four lines. Traits contributing most heavily to variation were proline, ascorbic acid, chloride, and sulfate ions, whereas those not making significant contributions were root length and shoot length (Figure 9). By excluding traits that did not add to the variation, it was possible to utilize samples from a broader set of sorghum lines. Except for seed vigor-II, all other traits contributing enormously to the first four principal components had estimates of precision associated with their measurement below 10%. Given that the first three principal components accounted for 92% of the variation in this population of samples, and given that germination traits, proline, ascorbic acid, antioxidants, cations and anions, osmotic potential, and total soluble sugars contributed most strongly to these principal components, a strong argument can be made that, in the practice of evaluating or screening sorghum for tolerance and adaptability under salt stress, emphasis should be placed on these traits. The effects of the different seed priming treatments on sorghum germination traits under salt stress were evaluated by principal component analysis (PCA) (Table 7). The significant effects of the seed priming treatments on germination traits were distributed along the PC1 axis in the order of SSG 59-3 > G-46 > CSV44F > PC-5 at 100 mM NaCl.

## 4. Discussion

Plants can tolerate salinity, but the extent to which they can counteract this menace depends on the nature of a species or even a cultivar [45]. Sorghum displays a significant intraspecific difference in salinity tolerance [7]. Seedling growth is a critical stage for the establishment of plant population under saline conditions. Salinity primarily reduces the osmotic ability of soil solutions to impede water intake by seed, which affects the seed germination rate. It inhibits seed germination by reducing the ease with which the water can be absorbed by the seeds or the Na^+^ and Cl^−^ ions [46] and even in the case of salt-tolerant plants. The germination index, dry leaf weight, and root length are the most significant factors and are recommended as the main indexes to identify alkaline tolerance of sorghum at germination [47]. Salinity stress drastically decreased the seedling vigor of screened sorghum genotypes due to an increase in salt concentration, which may cause delayed emergence of plumule and radical [9]. The growth rate of shoots controls the varietal difference in that lower shoot growth is associated with a higher level of NaCl in the leaves. Reduced shoot growth could also be due to reduced leaf initiation and progression and accelerated leaf abscission and internode growth [48]. The percent of root biomass showed a downward trend because the root was the first contacting organ to the treatment solution. This was demonstrated by the phenomenon that the root accumulated more Na^+^ and suffered more severe iron toxicity. This mechanism operates in salt-sensitive and salt-tolerant genotypes but with different efficiency levels [49]. The analysis revealed that germination rate, leaf weight, and root length, which were the most significant loads in the germination factors, leaf, and root, respectively, will be used as the main indexes to screen for sorghum salt tolerance. Progress has been made in identifying traits that are good indicators of salt tolerance in sorghum. Sun et al. [50] studied the responses of 42 sorghum genotypes to salt stress to identify genotypes with tolerance to salinity during germination.

Stress affects several physiological processes throughout the plasma membrane by impairing photosynthesis (especially PSII damage), resulting from inadequate water supply and its translocation together with ions and organic solutes. Stress also reduces the capacity of leaf osmosis [51]. Saline stress generally results in a reduction in chlorophyll and photosynthetic thresholds [7]. Total chlorophyll content was significantly affected by an increment of salinity levels, which showed that higher salt concentrations could ramp up chlorophyll decomposition in the body and reduce the photosynthetic efficiency of sorghum plants. In high salinity, the chlorophyll content initially increased and decreased afterward, suggesting that the optimal salt concentration may contribute to more chlorophyll synthesis, absorbing more light for photosynthesis [52].

The protective antioxidant system is crucial for a plant’s survival under stress conditions. Salt stress significantly impacts the lipid peroxidation of the plasma membrane, suggesting disruption and leakage of membranes under stressful situations [53]. MDA content increased due to the accumulation of reactive oxygen species (ROS) synthesized under stress conditions [54,55]. The increase in membrane lipid peroxidation during oxidative stress results from increased production of reactive oxygen species [56]. However, biological membrane stability has been considered an effective tool to evaluate the negative impacts of salinity stress. Similar to our results, low MDA content has been shown to be essential in salt tolerance, as other researchers represent. Salt-tolerant barley cultivar and the salt-resistant canola plant [57] also had lower lipid peroxidation levels, which is an essential hint of higher oxidative damage limiting growth capacity under salinity. However, salt-sensitive rice and maize varieties had higher MDA content and electrolyte leakage in response to salt stress [19]. It has been suggested that a decrease in membrane stability reflects the extent of lipid peroxidation caused by ROS [58].

Ascorbic acid is a common constituent of a plant cell synthesized in the cytosol and then translocated to the apoplast [59]. The primary function of ascorbate is to protect plant cells against oxidative damage by scavenging oxygen-free radicals directly. A higher level of ascorbic acid content in tolerant genotypes in the present study is in accordance with the observations of Almeselmani et al. [60], who reported higher ascorbic acid content in tolerant wheat genotypes. Compatible solutes, antioxidants, and soluble sugars constitute the biochemical basis of varietal differences associated with salt tolerance [2]. Total soluble carbohydrates act as an osmolyte inside the plant cell, and their accumulation increases the resistance against stress conditions [15]. Generally, the plant’s osmotic regulators include organic compounds and inorganic ions. In this study, with the increasing salt stress, proline and soluble carbohydrate content in leaves of sorghum seedlings also increased to mitigate oxidative stress. Proline and soluble sugar are the most common compounds that accumulate as compatible osmolytes [16]. Plants survive in saline conditions due to osmotic adjusting, including intracellular division and separation of toxic ions from the cytosol via energy-related transport into the vacuole survives [61,62]. The lower cytosolic Na^+^ ion and K^+^/Na^+^ homeostasis generally has been reported to be an essential for salinity tolerance, and higher concentrations of K^+^/Na^+^ have been suggested in the salt-tolerant lines [1]. A large proportion of Na^+^ translocates into the plant’s body under saline stress, affecting K^+^ absorption and the disruption of the equilibrium between Na^+^ and K^+^ [63]. With the increasing saline stress, the Mg^2+^ ion and Ca^2+^ ion in leaves of sorghum seedlings were decreased significantly. This showed that salt stress affected the Ca^2+^ and Mg^2+^ processes [64]. The Cl^-^ and SO_4_^2-^ anions showed a similarly increasing trend of increasing salinity, whereas the accumulation of Cl^-^ ions was significantly higher than the SO_4_^2-^ ion. With the reduced concentration of Ca^2+^ ions, the function of the cell membrane is disrupted, which maintains the integrity and regulates the selective transport of Na^+^ and K^+^ ions [24].

Starch is the most abundant storage carbohydrate and the primary energy supplier [6,16]. Wang et al. [65] found that starch content in sorghum genotypes ranged between 64 and 74% of dry grain weight. SSG 59-3 synthesized maximum proteins after exposure to salt treatment, whereas it was slower in PC-5. Thus, germinating seeds of SSG 59-3 showed higher starch mobilization and enhanced de novo protein synthesis at higher salinity levels [18]. The starch granules are surrounded by a protein matrix that can limit the access of enzymes [66]. Thus, its utilization depends upon the genetic make-up of cultivars for its effective mobilization [67]. As starch comprises about 71.2% of reserve food in sorghum seed, it is the primary energy source for germinating seeds [14].

In the present study, attempts were made to establish a correlation between antioxidative defense mechanisms and salinity-induced changes in sorghum seedlings. During oxidative stress, the excess production of ROS is scavenged by a complex enzymatic antioxidative system, which controls ROS production and ultimately protects the plant against oxidative damage [15]. This salinity-induced defense mechanism is differential and primarily dependent on differential antioxidant enzymes, salinity extent, and exposure time [68]. SOD is the most effective intracellular enzymatic antioxidant, ubiquitous in all aerobic organisms and all subcellular compartments prone to ROS-mediated oxidative stress [2,10]. Lee et al. [69] reported that transgenic tobacco plants overexpressing Cu/Zn-SOD showed tolerance to salt and water stresses. The induction of catalase activity was reported on the accumulation of H_2_O_2_ and is seemingly consistent with this enzyme’s role in scavenging enhanced H_2_O_2_ levels [24]. Salinity-induced stimulation in POD activity in tolerant genotypes, suggesting their possible role in efficiently removing H_2_O_2_ in tolerant genotypes [70]. Enhanced peroxidase activity under various stresses was linked to protection from oxidative damage, lignification, and cross-linking of the cell wall. APX and GPX are specific enzymes that scavenge chloroplastic H_2_O_2_ using ascorbate as an electron donor in the first step of the ascorbate–glutathione cycle and are considered the essential plant peroxidases in H_2_O_2_ detoxification [71]. The enhanced activity of APX and GPX concomitant with an enhanced ascorbic acid content and glutathione may help satisfy ROS [72]. GR is essential to recycle GSH in the ascorbate–glutathione cycle in an NADPH-dependent reaction. Similarly, a higher induction in GR activity in tolerant varieties than susceptible varieties was reported in *Macrotyloma uniflorum* and chickpea [73]. Similar results were also reported in maize nine and wheat differing in salt tolerance [74].

Several genes have been described to play an essential role in numerous physiological processes of embryo development, germination, and maturation [75]. During the germination process, the genes involved are catabolism or α-amylase synthesis and degradation. Under high salinity, there is rapid mobilization of reserved food stored [18]. The scutellum is the initial site of *α-amy1* expression and subsequent α-amylase synthesis. GBSS plays a minor role in amylose synthesis in non-storage tissue [76]. The differential expression of ion transporter genes under different salinity levels indicated their adaptive behavior for Na^+^ ion exclusion. Salt stress elicits a cytosolic calcium signal [77]. The increased intracellular concentration of Ca^2+^ ions is sensed by *SOS3,* which interacts and activates *SOS2*, a serine/threonine-protein kinase [78]. Both *SOS2* and *SOS3* regulate the expression level of *SOS1*, a salt tolerance gene that encodes a sodium proton antiporter (*NHX*) [79]. Co-expression of *H^+^-PPase* together with *NHX* dramatically enhances the salt tolerance capacity in plants [80]. One of the largest families of plant-specific transcription factors includes *NAC* regulating several other genes’ transcription in response to abiotic stress [2]. Overexpression of *SbVPPase* in transgenic finger millet enhances the plant’s performance under salt stress [81]. Sun et al. [82] elucidated that the complex regulatory network involving non-coding RNAs underpin the sorghum tolerance to salt stress. Varoquaux et al. [83] assessed the molecular underpinnings of drought responses in sorghum and reported that, under stressed conditions, massive changes occurred in the transcriptome, impacting more than 40% of all expressed genes encompassing a wide variety of molecular mechanisms.

The principal component analysis is a multivariate technique for examining the relationships among several quantitative variables [84]. Principal component analysis (PCA) provides mechanisms to describe relationships between the germination potential and adaptive means of sorghum seedlings under high salinity [85]. This tool had led us to identify factors that will be the focus on future efforts to perform targeted changes to affect sorghum adaptability and tolerance behavior positively. Analysis of variability among traits and knowledge of associations among traits contributing to yield would be of great importance in planning a successful breeding program [72]. Abraha et al. [86] reported four principal components with eigenvalues greater than one, which explained > 75% of the total variation for agronomic traits.

## 5. Conclusions

The present study has presented a comprehensive account of past developments and current trends related to antioxidative changes at the transcriptomic level and the instigation of molecular interactions, as well as their possible collaboration with one another in sorghum under salt stress. Thresholds of sorghum-specific antioxidant defense mechanisms are stimulated at early growth stages. The rapid mobilization of starch and proteins in tolerant genotypes might be due to increased α-amylase and protease activity and de novo protein synthesis. Upregulation of Na^+^/K^+^ transporters provided an in-depth analysis of the Na^+^ exclusion mechanism to prevent ion toxicity. It can be concluded that seedling traits can be used as valid criteria for the selection of genotypes with better tolerance to salinity stress. Deciphering these specific alternatives could help to develop more efficient metabolic engineering mechanisms specific to different organs and ages to cope with particular stress conditions. The information gained from this study might help to establish a salt-responsive sorghum network and provide insights and directions for salt-tolerant germplasm maintenance and cereal crop improvement. The study highlights the need for further understanding and exploration of plant sensing and signaling systems.

## Figures and Tables

**Figure 1 plants-10-02463-f001:**
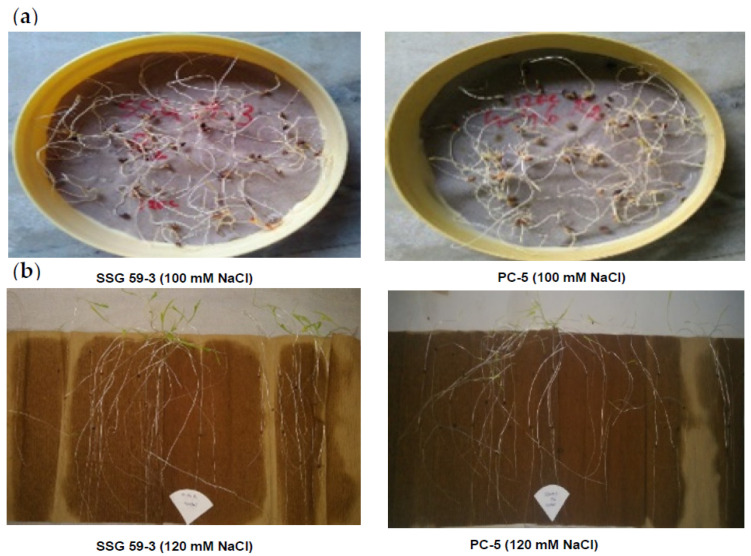
Germination experiment in sorghum genotypes under salt stress. (**a**): Petri plate experiment; (**b**): between-paper experiment.

**Figure 2 plants-10-02463-f002:**
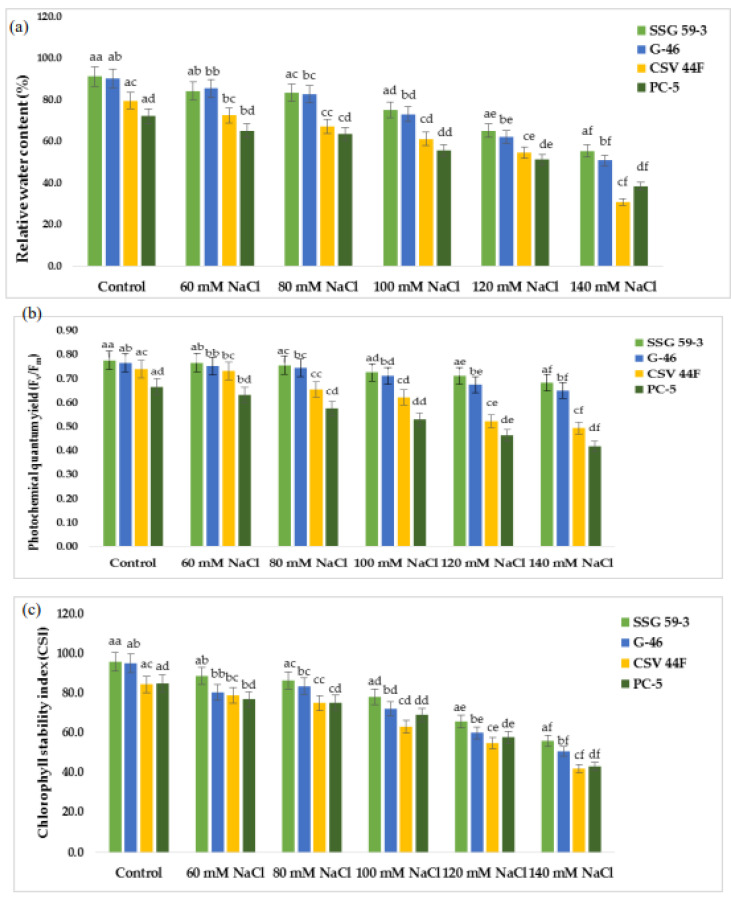
Effect of salt stress on (**a**) relative water content, (**b**) photochemical quantum yield (F_v_/F_m_), and (**c**) chlorophyll stability index of sorghum genotypes. ^a–f^ Values with different superscripts in the same row are significantly different at *p* < 0.05.

**Figure 3 plants-10-02463-f003:**
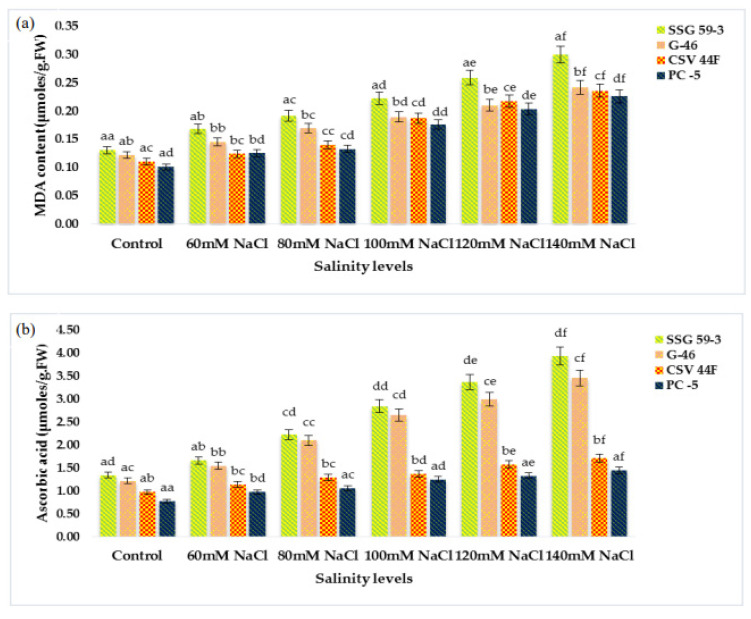
Effect of salinity on (**a**) MDA, (**b**) ascorbic acid, (**c**) proline, and (**d**) total soluble carbohydrates of sorghum seedlings under saline stress. Values represent means ± S.E. ^a–f^ Values with different superscripts in the same row are significantly different at *p* < 0.05.

**Figure 4 plants-10-02463-f004:**
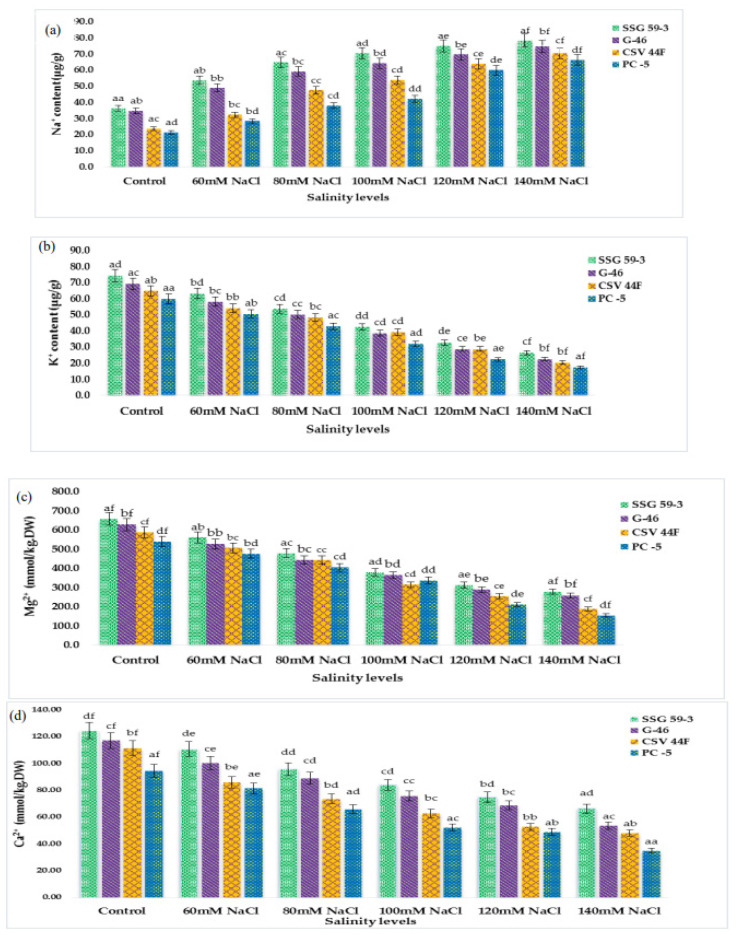
Accumulation of inorganic cations and anions of sorghum seedlings due to oxidative stress. (**a**) Na^+^ content; (**b**) K^+^ content; (**c**) Mg^2+^ content; (**d**) Ca^2+^ content; (**e**) Cl^−^ content; (**f**) SO_4_^2−^ content. Values represent means ± S.E. ^a–f^ Values with different superscripts in the same row are significantly different at *p* < 0.05.

**Figure 5 plants-10-02463-f005:**
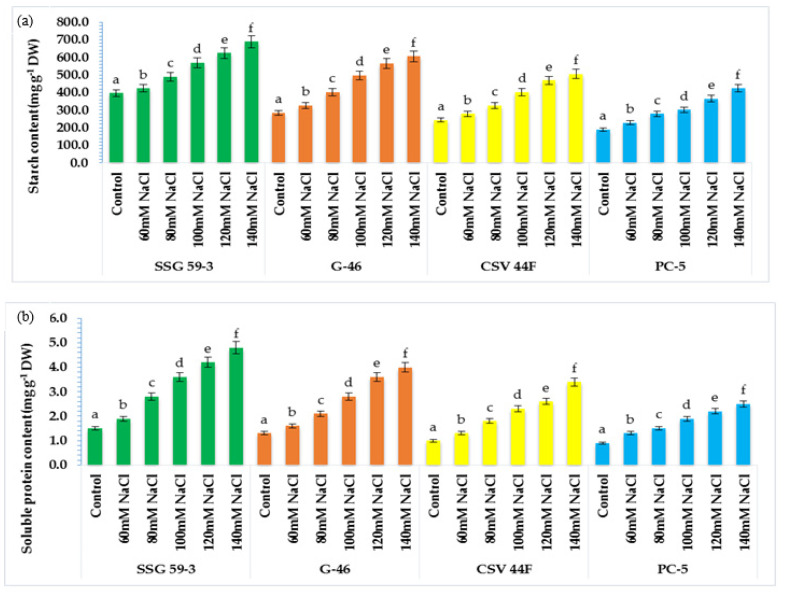
Effect of salinity on (**a**) starch and (**b**) protein content in germinating seeds of sorghum. Each value is the mean of triplicate. ^a–f^ Values with different superscripts in the same row are significantly different at *p* < 0.05.

**Figure 6 plants-10-02463-f006:**
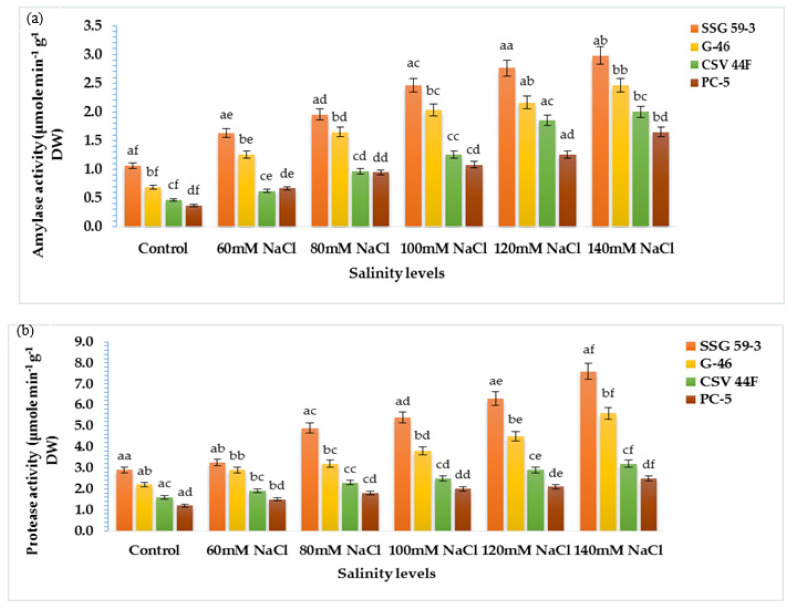
Effect of salinity on (**a**) α-amylase and (**b**) protease activity in germinating sorghum seedlings. Each value is the mean of triplicate. ^a–f^ Values with different superscripts in the same row are significantly different at *p* < 0.05.

**Figure 7 plants-10-02463-f007:**
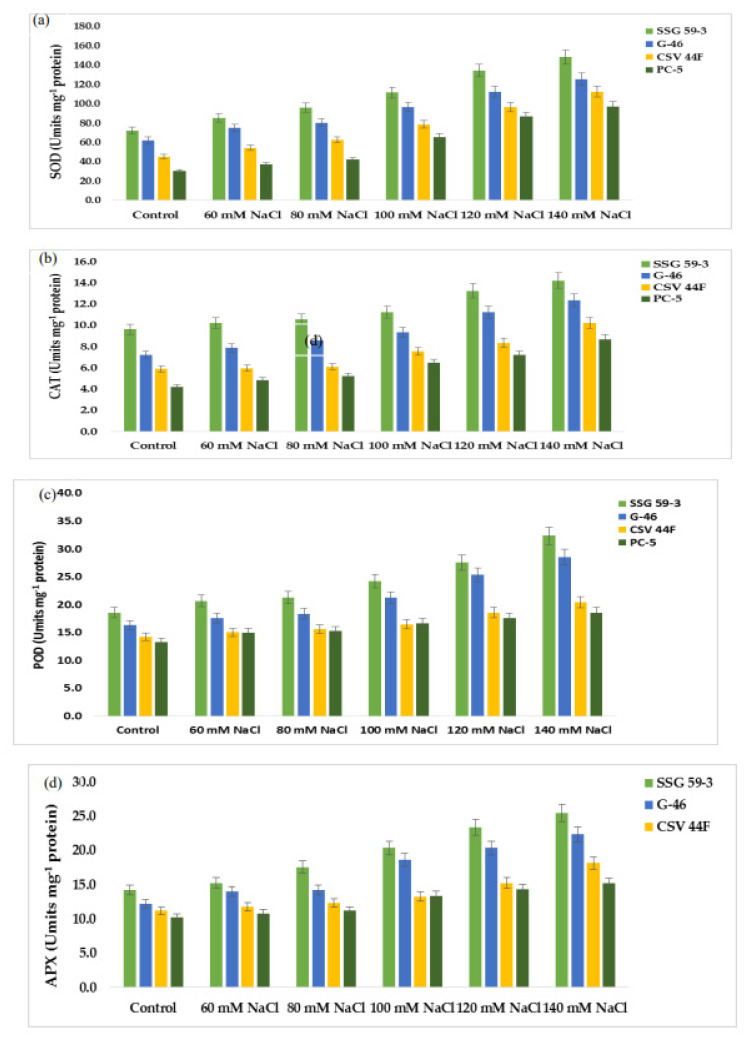
Effect of salt stress on (**a**) superoxide dismutase (SOD), (**b**) catalase (CAT), (**c**) peroxidase (POD), (**d**) ascorbate peroxidase (APX), (**e**) glutathione peroxidase (GPX), and (**f**) glutathione reductase (GR) of sorghum genotypes. Values are means of at least three replicates and significant differences between means, as determined by Tukey’s test (*p* < 0.05).

**Figure 8 plants-10-02463-f008:**
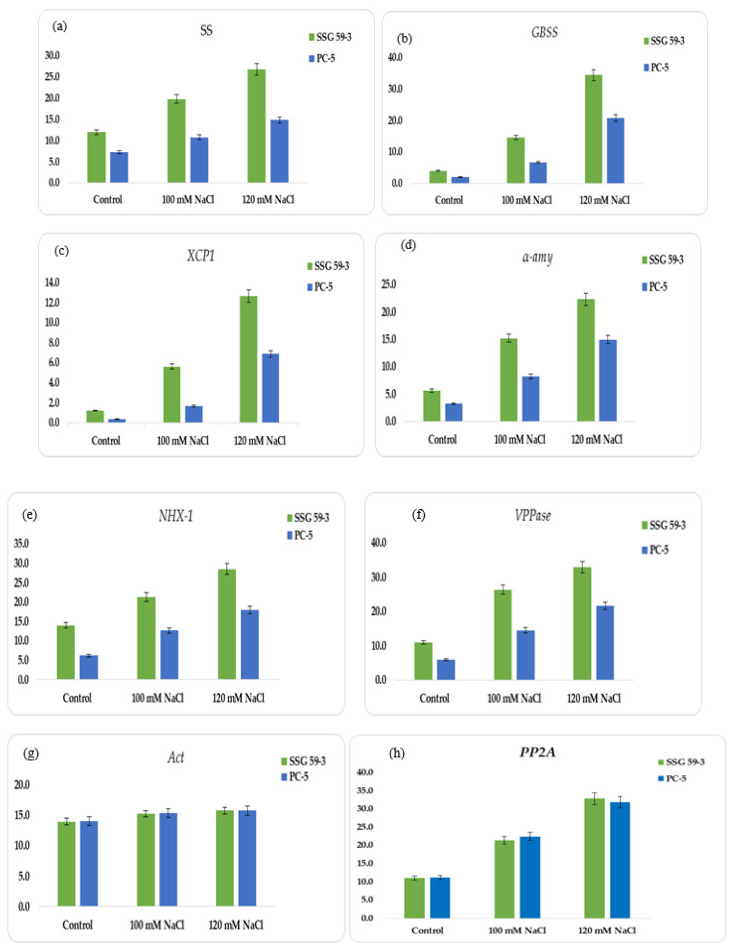
Relative quantification of: (**a**) Sucrose synthase (*SS*), (**b**) granule-bound starch synthase (*GBSS*), (**c**) cysteine protease (*XCP1*), (**d**) α-amylase synthesis (*α-amy*), (**e**) sodium proton antiporter (*NHX-1*), (**f**) vacuolar-proton pyrophosphatase (*VPPase-I*), (**g**) actin (*Act*), and (**h**) *PP2A*.

**Figure 9 plants-10-02463-f009:**
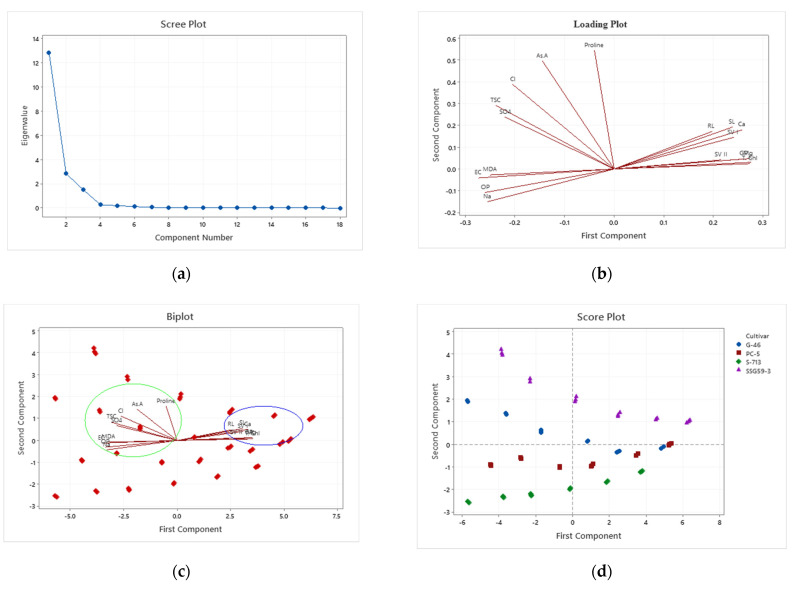
Plots of principal components of four sorghum lines. (**a**) scree plot; (**b**) loading plot; (**c**) biplot; (**d**) score plot.

**Table 1 plants-10-02463-t001:** Physical characteristics of sorghum genotypes.

Cultivars	Description	Source	Pedigree
SSG 59-3	Sweet Sudan Grass	CCS HAU Hisar, India	Non-sweet sudangrass × JS-263
G-46	Germplasm 46	CCS HAU Hisar, India	Selection from S-202 which is a selection from cross 10626B × 6090 M3-1-1
PC-5	Pant Chari 5	Pantnagar, India	CS 3541 × IS 6935
CSV 44F	Haryana Jowar	CCS HAU, Hisar, India	Hybrid 308 × 437-1

**Table 2 plants-10-02463-t002:** List of primers.

Gene	Accession No.	Primers	Sequence (5′-3′)
*α-amy*	XM_002452836	Forward	CTGGTATGATTGGGTGGTGG
		Reverse	CATATGATCCAGTGTGGCCC
*GBSS*	AF079258	Forward	GGCAAGAAGAAGTTCGAGCG
		Reverse	TCAGCAATAGGAGCGAGACA
*SS*	AF168786	Forward	CCATGCCGCTGAACAATACG
		Reverse	ACACCATGCGCTTTACAGGA
*XCP1*	XM_002439083	Forward	AGTCGTTTCAGGTACGGTGG
		Reverse	CATGCGGATGTAGCCCTTCT
*NHX-1*	EU482408.2	Forward	CGATGGGTGAACGAGTCCAT
		Reverse	GTTGCAAAAGTATGTCTGGCA
*VPPase-I*	GQ469975.1	Forward	CACCTCTCTGGTATCTGGTTTC
		Reverse	GTGCGGGCTCAATTTCTTTC
*Act 1*	NC_012870.2	Forward	CGCTGGCATACAGAGAGAGG
		Reverse	CCCGAGCTTTGCTGTAGGTA

**Table 3 plants-10-02463-t003:** Effect of salinity on germination indices of sorghum seedlings during the early growth stage.

Genotypes		SSG 59-3	G-46	CSV 44F	PC-5
Germination Percent (%)					
	Control	95.98 ± 0.97 ^a,d^	93.98 ± 0.95 ^a,c^	86.11 ± 0.86 ^a,b^	79.91 ± 0.92 ^a,a^
	60 mM NaCl	94.36 ± 0.54 ^a,d^	89.06 ± 1.05 ^b,c^	72.99 ± 0.74 ^b,b^	68.30 ± 0.68 ^a,b^
	80 mM NaCl	80.62 ± 1.21 ^c,d^	74.96 ± 1.16 ^c,c^	64.41 ± 0.76 ^b,b^	58.60 ± 0.33 ^a,b^
	100 mM NaCl	73.73 ± 1.12 ^d,d^	66.33 ± 0.66 ^c,d^	51.66 ± 0.80 ^b,d^	45.53 ± 0.46 ^a,b^
	120 mM NaCl	64.41 ± 0.76 ^d,e^	56.18 ± 0.33 ^c,e^	43.56 ± 0.44 ^b,e^	38.13 ± 0.22 ^a,e^
	140 mM NaCl	54.70 ± 0.85 ^d,f^	48.51 ± 0.49 ^c,f^	34.53 ± 0.40 ^b,f^	29.10 ± 0.17 ^a,f^
**Root Length (cm)**					
	Control	14.50 ± 0.15 ^d,f^	12.50 ± 0.13 ^c,f^	8.12 ± 0.08 ^b,f^	7.69 ± 0.09 ^a,f^
	60 mM NaCl	12.42 ± 0.07 ^d,e^	11.68 ± 0.13 ^c,e^	6.40 ± 0.07 ^b,e^	5.54 ± 0.06 ^a,e^
	80 mM NaCl	11.31 ± 0.17 ^d,d^	10.53 ± 0.16 ^c,d^	5.34 ± 0.06 ^b,d^	4.27 ± 0.02 ^a,d^
	100 mM NaCl	10.32 ± 0.15 ^c,d^	9.31 ± 0.10 ^c,c^	4.66 ± 0.07 ^b,c^	3.56 ± 0.04 ^a,c^
	120 mM NaCl	9.66 ± 0.11 ^b,d^	8.73 ± 0.05 ^b,c^	3.27 ± 0.04 ^b,b^	2.91 ± 0.02 ^a,b^
	140 mM NaCl	8.41 ± 0.13 ^a,d^	7.03 ± 0.07 ^a,c^	1.68 ± 0.02 ^a,b^	1.01 ± 0.01 ^a,a^
**Shoot Length (cm)**					
	Control	22.50 ± 0.23 ^d,f^	21.29 ± 0.22 ^c,f^	16.28 ± 0.18 ^b,f^	14.35 ± 0.15 ^a,f^
	60 mM NaCl	20.66 ± 0.12 ^d,f^	20.33 ± 0.24 ^c,e^	12.87 ± 0.13 ^b,e^	11.40 ± 0.12 ^a,e^
	80 mM NaCl	16.22 ± 0.24 ^d,d^	15.40 ± 0.24 ^c,d^	11.43 ± 0.06 ^b,d^	9.66 ± 0.11 ^a,d^
	100 mM NaCl	14.26 ± 0.21 ^c,d^	13.17 ± 0.13 ^c,c^	9.90 ± 0.10 ^b,c^	6.48 ± 0.10 ^a,c^
	120 mM NaCl	12.07 ± 0.13 ^b,d^	12.24 ± 0.07 ^b,c^	9.53 ± 0.06 ^b,c^	5.35 ± 0.06 ^a,b^
	140 mM NaCl	11.56 ± 0.19 ^a,d^	10.59 ± 0.11 ^a,c^	7.93 ± 0.05 ^a,b^	3.21 ± 0.12 ^a,a^
**Seed Vigor I**					
	Control	4109.31 ± 41.51 ^d,f^	3733.37 ± 37.72 ^c,f^	2672.55 ± 26.59 ^a,f^	2837.27 ± 32.55 ^b,f^
	60 mM NaCl	3460.16 ± 19.91 ^d,e^	3075.06 ± 35.99 ^c,e^	2050.66 ± 20.72 ^a,e^	2051.16 ± 11.80 ^b,d^
	80 mM NaCl	3267.07 ± 49.21 ^d,d^	2918.17 ± 45.07 ^c,d^	2123.51 ± 24.85 ^a,d^	1842.57 ± 18.34 ^b,e^
	100 mM NaCl	2908.69 ± 44.16 ^c,d^	2717.37 ± 26.91 ^c,c^	1520.37 ± 23.48 ^a,c^	1558.00 ± 15.50 ^b,c^
	120 mM NaCl	2425.43 ± 28.38 ^b,d^	2152.01 ± 12.51 ^b,c^	1151.29 ± 11.40 ^a,b^	1324.31 ± 7.70 ^b,b^
	140 mM NaCl	2140.77 ± 33.07 ^a,d^	1106.25 ± 10.96 ^a,c^	490.31 ± 5.62 ^a,a^	409.85 ± 2.37 ^a,b^
**Seed Vigor II**					
	Control	19.66 ± 0.20 ^c,e^	25.33 ± 0.26 ^d,e^	24.09 ± 0.24 ^b,e^	22.36 ± 0.26 ^a,f^
	60 mM NaCl	26.82 ± 0.16 ^c,f^	30.19 ± 0.35 ^d,f^	24.00 ± 0.24 ^b,e^	19.47 ± 0.20 ^a,d^
	80 mM NaCl	21.63 ± 0.33 ^c,d^	19.25 ± 0.30 ^d,d^	19.46 ± 0.23 ^b,d^	19.20 ± 0.11 ^a,d^
	100 mM NaCl	19.00 ± 0.29 ^c,e^	19.80 ± 0.20 ^c,d^	12.15 ± 0.19 ^b,c^	14.52 ± 0.15 ^a,c^
	120 mM NaCl	17.11 ± 0.20 ^b,c^	19.40 ± 0.12 ^d,d^	10.89 ± 0.11 ^b,b^	10.70 ± 0.06 ^a,b^
	140 mM NaCl	19.92 ± 0.31 ^c,e^	19.80 ± 0.20 ^c,d^	4.60 ± 0.06 ^a,b^	4.01 ± 0.02 ^a,a^

All analyzed data are expressed as mean ± SD. Values are presented on a fresh weight basis. ^a–f^ Values with different superscripts in the same row are significantly different at *p* < 0.05.

**Table 4 plants-10-02463-t004:** The physiological indices of sorghum seedlings under saline conditions.

	Osmotic Potential (-MPa)				Total Chlorophyll (mg/g DW)			
Genotype	SSG 59-3	G-46	CSV 44F	PC-5	SSG 59-3	G-46	CSV 44F	PC-5
Control	0.63 ± 0.006 ^a,a^	0.68 ± 0.008 ^a,b^	0.76 ± 0.008 ^a,c^	0.86 ± 0.009 ^a,d^	3.32 ± 0.034 ^a,d^	3.23 ± 0.037 ^a,c^	2.97 ± 0.030 ^a,b^	2.75 ± 0.028 ^a,a^
60 mM NaCl	0.77 ± 0.005 ^a,b^	0.82 ± 0.009 ^b,b^	0.95 ± 0.011 ^b,c^	0.95 ± 0.010 ^b,d^	2.78 ± 0.016 ^b,d^	2.71 ± 0.027 ^b,c^	2.85 ± 0.033 ^b,b^	2.54 ± 0.026 ^a,b^
80 mM NaCl	0.86 ± 0.013 ^a,c^	0.83 ± 0.005 ^b,b^	1.27 ± 0.020 ^c,c^	1.33 ± 0.016 ^c,d^	2.61 ± 0.039 ^c,d^	2.52 ± 0.014 ^b,c^	2.52 ± 0.039 ^b,c^	2.33 ± 0.027 ^a,c^
100 mM NaCl	1.17 ± 0.018 ^a,d^	1.23 ± 0.013 ^b,c^	1.35 ± 0.014 ^c,d^	1.44 ± 0.022 ^d,d^	2.28 ± 0.035 ^d,d^	2.15 ± 0.021 ^c,d^	2.02 ± 0.020 ^b,d^	1.89 ± 0.029 ^a,d^
120 mM NaCl	1.30 ± 0.016 ^a,e^	1.30 ± 0.008 ^b,d^	1.48 ± 0.009 ^c,e^	1.52 ± 0.015 ^d,e^	1.88 ± 0.022 ^d,e^	1.79 ± 0.010 ^c,e^	1.77 ± 0.010 ^b,e^	1.66 ± 0.016 ^a,e^
140 mM NaCl	1.38 ± 0.021 ^a,e^	1.42 ± 0.008 ^b,e^	1.56 ± 0.016 ^c,f^	1.62 ± 0.018 ^d,f^	1.77 ± 0.028 ^d,f^	1.63 ± 0.009 ^c,f^	1.52 ± 0.015 ^b,f^	1.41 ± 0.016 ^a,f^

All analyzed data are expressed as mean ± SD. Values are presented on a fresh weight basis. ^a–f^ Values with different superscripts in the same row are significantly different at *p* < 0.05.

**Table 5 plants-10-02463-t005:** Membrane injury in sorghum seedlings under salt stress.

Membrane Injury (%)				
	SSG 59-3	G-46	PC-5	CSV 44F
Control	16.54 ± 0.17 ^a,a^	18.31 ± 0.21 ^a,b^	20.33 ± 0.21 ^a,c^	23.31 ± 0.23 ^a,d^
60 mM NaCl	22.29 ± 0.13 ^a,b^	24.53 ± 0.24 ^b,b^	28.62 ± 0.33 ^b,c^	29.90 ± 0.30 ^b,d^
80 mM NaCl	29.93 ± 0.45 ^a,c^	33.64 ± 0.19 ^b,c^	35.11 ± 0.54 ^c,c^	37.91 ± 0.44 ^c,d^
100 mM NaCl	35.82 ± 0.54 ^a,d^	39.38 ± 0.39 ^b,d^	39.03 ± 0.39 ^c,d^	43.22 ± 0.67 ^d,d^
120 mM NaCl	41.48 ± 0.48 ^a,e^	43.37 ± 0.25 ^b,e^	44.92 ± 0.26 ^c,e^	47.16 ± 0.47 ^d,d^
140 mM NaCl	46.57 ± 0.72 ^a,f^	47.99 ± 0.28 ^b,e^	52.02 ± 0.52 ^c,f^	53.78 ± 0.62 ^d,e^

All analyzed data are expressed as mean ± SD. Values are presented on a fresh weight basis. ^a–f^ Values with different superscripts in the same row are significantly different at *p* < 0.05.

**Table 6 plants-10-02463-t006:** Eigenanalysis of the covariance matrix using four sorghum lines and eighteen variables.

	Eigenvalue	Proportion	Cumulative
Principal component 1	12.86	0.72	0.72
Principal component 2	2.84	0.16	0.87
Principal component 3	1.51	0.08	0.96
Principal component 4	0.27	0.02	0.97
Principal component 5	0.19	0.01	0.98
Principal component 6	0.12	0.01	0.99
Principal component 7	0.05	0.00	0.99
Principal component 8	0.05	0.00	0.99
Principal component 9	0.03	0.00	1.00
Principal component 10	0.02	0.00	1.00
Principal component 11	0.02	0.00	1.00
Principal component 12	0.01	0.00	1.00
Principal component 13	0.01	0	1.00
Principal component 14	0.01	0	1
Principal component 15	0.00	0	1
Principal component 16	0.00	0	1
Principal component 17	0.00	0	1
Principal component 18	0	0	1

**Table 7 plants-10-02463-t007:** Eigenvectors of the principal components using four sorghum lines and eighteen variables.

	*PC1	PC 2	PC 3	PC 4	PC 5	PC 6	PC 7	PC 8	PC 9	PC 10
K^+^	0.28	0.03	0.09	−0.02	0.10	−0.06	0.25	−0.03	0.16	−0.02
Na^+^	−0.26	−0.15	−0.16	0.07	−0.42	−0.22	0.41	0.01	−0.43	0.00
Mg^2+^	0.27	0.05	0.13	0.02	−0.01	−0.18	0.37	−0.24	0.15	0.03
Ca^2+^	0.26	0.18	0.13	−0.02	0.27	0.04	−0.01	−0.05	0.03	0.03
SO_4_^2−^	−0.22	0.24	−0.29	0.10	0.53	0.33	0.45	−0.08	−0.10	−0.23
Cl^−^	−0.20	0.39	−0.01	0.06	−0.18	−0.21	0.06	−0.63	−0.15	−0.02
EC	−0.27	−0.04	−0.11	0.00	−0.09	0.05	0.02	0.22	−0.08	0.50
OP	−0.26	−0.11	−0.20	−0.17	−0.23	−0.06	0.17	−0.07	0.68	−0.11
TCC	0.27	0.03	0.09	0.01	0.02	−0.20	0.43	0.01	0.17	0.50
TSC	−0.24	0.29	−0.07	−0.13	0.19	0.05	−0.09	−0.11	0.00	0.34
MDA	−0.25	−0.03	−0.31	0.07	0.26	−0.48	−0.13	0.10	0.34	−0.11
AA	−0.14	0.50	0.12	0.03	−0.11	0.13	0.01	0.35	0.16	0.29
Proline	−0.04	0.55			−0.29	−0.11	−0.08	0.17	0.09	−0.33
GP	0.27	0.11			−0.22	0.14	−0.27	−0.34	0.11	0.07
RL	−0.24	−0.04	−0.50	−0.34	−0.08	0.10	−0.16	−0.14	−0.05	0.18
SL	0.24	0.19	−0.27	−0.07	0.17	−0.60	−0.10	0.25	−0.26	−0.04
SV-I	0.24	0.15	−0.30	−0.32	−0.26	0.22	0.28	0.33	−0.02	−0.27
SV-II	0.22	0.04	−0.37	0.83	−0.14	0.13	−0.04	0.07	0.10	0.06

* PC: Principal component; TSC: total soluble carbohydrates; TCC: total chlorophyll content; SV: Seed vigor; RL: root length; SL: shoot length; EC: electrical conductivity; OP: osmotic potential; AA: ascorbic acid; GP: germination percentage.

## Data Availability

Data are contained within the article or Appendix A.

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
