# Peer review of "Deciphering Reserve Mobilization, Antioxidant Potential, and Expression Analysis of Starch Synthesis in Sorghum Seedlings under Salt Stress"

_plants, 2021, doi:10.3390/plants10112463_

Round 1
Reviewer 1 Report
The original article entitled "Deciphering reserve food mobilization, antioxidant potential, and expression profiling of starch synthesizing genes in sorghum [Sorghum bicolor (L.) Moench] seedlings under salt stress" might deserve publication after some revisions and suggestions as reported below:
Title: Synthesize and improve;
Line 28: I would change "deciphered" to highlighted;
Lines 112-114/151-152/156-157/170-171: Write the formulas well with the equation function or alternatively as an image in order to make them clear and legible;
Line 536: Add “o” to score;
Figures: Improve the resolution of all figures. Do not restrict the figures because they are not very legible, but reduce them from the corner. Not is clear what kind of statistical analysis was used in the data analysis. Was only the standard deviation done? And how was the p value calculated?
Figure 2A: Check the legend;
Figure 5: I recommend changing color for each different cultivars;
Figure 9: separating the various plots into different or not side by side figures;
Table 6: Are there only 10 variables?
I recommend adding the article by Dehnavi et al. 2020 (Effect of Salinity on Seed Germination and Seedling Development of Sorghum (Sorghum bicolor (L.) Moench) Genotypes).
In my opinion I would summarize the discussion as it seems there are some repetitions.
Author Response
Pointwise answers to reviewer’s comments
English language and style are needful done as highlighted and minor spell check is also needful done.
|
S.No |
Comment |
Reply |
|
|
Reviewer #1: |
|
|
1 |
Title: Synthesize and improve; · |
The title has been improved.
Kindly refer line no. 2-3. |
|
2 |
Line 28: I would change "deciphered" to highlighted |
The word has been "deciphered" to has been replaced with “highlighted”.
Kindly refer line no. 38.
|
|
3 |
Lines 112-114/151-152/156-157/170-171: Write the formulas well with the equation function or alternatively as an image in order to make them clear and legible; |
The formulas have been written using the equation function.
Kindly refer line no. 324-326, 391-392, 404-405. |
|
4 |
Line 536: Add “o” to score; |
Added “o” to score.
Kindly refer line no. 1162. |
|
5 |
Figures: Improve the resolution of all figures. Do not restrict the figures because they are not very legible, but reduce them from the corner. Not is clear what kind of statistical analysis was used in the data analysis. Was only the standard deviation done? And how was the p value calculated? · |
The resolution of all the figures has been improved throughout the manuscript as per the suggestions.
Two-way ANOVA was conducted to check the significance of main effects (genotypes and salinity) and their interaction on growth indices followed by posthoc comparison (Tuckey's test) at 5% level (P≤0.05) using SPSS v23.0 software (SPSS for Windows, Chicago, IL, USA). |
|
6 |
Figure 2A: Check the legend; |
Rectified the mistake. New figure no. 3A.
Kindly refer line no. 776-777. |
|
7 |
Figure 5: I recommend changing color for each different cultivars; |
The colour of each cultivar has been changed as per the suggestions. New figure no. 6.
Kindly refer line no. 938-954 |
|
8 |
Figure 9: separating the various plots into different or not side by side figures; |
All the plots are designated to PCA, so they were not separated into different figures. |
|
9 |
Table 6: Are there only 10 variables? |
No, the PCA was calculated among the antioxidant metabolites and seed quality traits. The antioxidative enzymes were included for PCA. |
|
10 |
I recommend adding the article by Dehnavi et al. 2020 (Effect of Salinity on Seed Germination and Seedling Development of Sorghum (Sorghum bicolor (L.) Moench) Genotypes). |
The article Dehnavi et al. 2020 has been added in the text as reference no [1].
Kindly refer line no. 93. |
|
11 |
In my opinion I would summarize the discussion as it seems there are some repetitions. |
The repetitions have been deleted in the discussion section.
Kindly refer line no. 1194-1391. |

Reviewer 2 Report
The topic sounds good but authors did not collect all the information about this topic and the salt stress responses of sorghum species. The English language of this manuscript needs drastic revision and I recommend to authors to ask a native English speaker to revise this MS. Other big problem is that readers could not get enough information about the applied sorghum species and I think that these sorghum species are different, eg. sudangrass is Sorghum sudanese and not Sorghum bicolor as authors wrote in the title. I recommend this manuscript to be drastically revisioned and structured again supplied with current results from new articles.
Author Response
Pointwise answers to reviewer’s comments
English language editing has been done by using MDPI English editing services (submission id: 36145).
|
S.No |
Comment |
Reply |
|
|
Reviewer #2: |
|
|
1 |
· Does the introduction provide sufficient background and include all relevant references? Must be improved |
The introduction has been rewritten as per the suggestions.
Kindly refer line no. 91-266. |
|
2 |
Is the research design appropriate? Must be improved |
Needful done.
Kindly refer line no. 267-314. |
|
3 |
Are the methods adequately described? Can be improved |
The methods have been improved as suggested.
Kindly refer line no. 315-628. |
|
4 |
Are the results clearly presented? Must be improved |
The results have been improved as suggested.
Kindly refer line no. 661-1193. |
|
5 |
· Are the conclusions supported by the results? Must be improved |
The conclusion has been rewritten to clearly present the outcomes of the study.
Kindly refer line no. 1392-1408. |
|
6 |
The topic sounds good but authors did not collect all the information about this topic and the salt stress responses of sorghum species. The English language of this manuscript needs drastic revision and I recommend to authors to ask a native English speaker to revise this MS. Other big problem is that readers could not get enough information about the applied sorghum species and I think that these sorghum species are different, eg. sudangrass is c and not Sorghum bicolor as authors wrote in the title. I recommend this manuscript to be drastically revisioned and structured again supplied with current results from new articles. |
The English language of this manuscript has been revised by using mdpi English editing services (submitted ID: 36145). The latest references have been added in the introduction and discussion part. According to the pedigree of SSG 59-3, it is Non-sweet sudangrass x JS-263. The information about the pedigree of the studied genotypes has been provided by the breeders of Forage section, Generics & Plant Breeding they suggested that it is Sorghum bicolor not Sorghum bicolor.
Kindly refer to Table no 1, line no. 283. |

Round 2
Reviewer 1 Report
I thank the authors for improving the manuscript by following all the suggestions of the reviewer. The only thing I would suggest improving is the solving of the equations. Therefore, in my opinion the Article "Deciphering reserve mobilization, antioxidant potential, and expression analysis of starch synthesis in sorghum seedlings under salt stress" deserves to be accepted for publication after minor revision
Reviewer 2 Report
Authors improved earlier version of manuscript, now it has good quality for this journal.